# 🐋MARINEMAID: DATASET AND BENCHMARK ON DETECTING AND UNDERSTANDING MARINE CREATURES

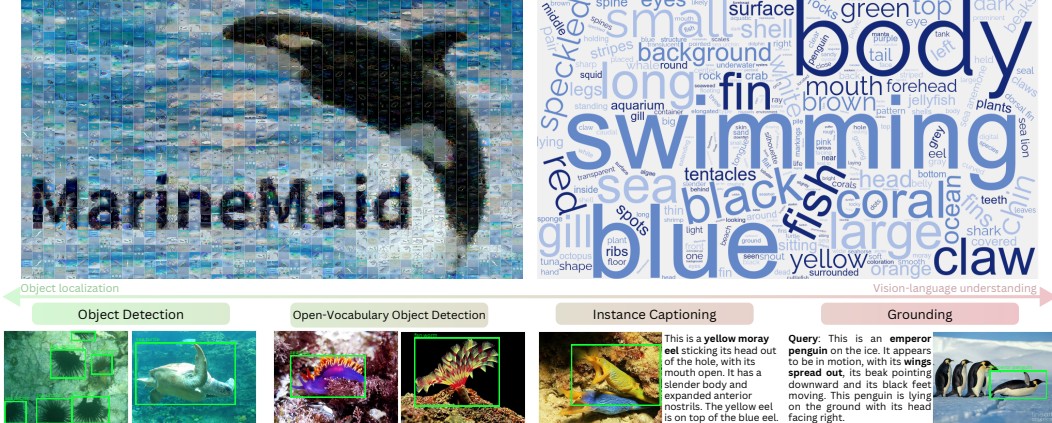

Figure 1: **We present MarineMaid dataset, the first dataset and benchmark specifically designed for marine visual understanding.** *Top:* MarineMaid consists of two main components: high-quality bounding boxes for object detection and fine-grained instance captions for marine vision-language understanding. *Bottom:* MarineMaid enables a wide range of marine visual understanding tasks, including *object detection*, *open-vocabulary object detection*, *instance captioning* and *grounding*.

## ABSTRACT

Oceans, covering more than 70% surfaces of our blue planets are less explored by the whole computer vision community. The scarcity of the labeled data is attributed to the most hindering issue. In this work, we propose a novel and comprehensive dataset called **MarineMaid** specifically designed for marine monitoring and understanding, including a wide spectrum of marine creatures. Based on the essential requirements of the marine research community, we adopt *object detection* and *vision-language understanding* as our two fundamental tasks. The former object detection could yield precise localization and category predictions for species identification and monitoring. Besides the sole category and BBOX predictions, the latter vision-language understanding generates redundant and comprehensive captions about biological traits required for domain experts. MarineMaid contains **12,873** fine-grained instance-captioning pairs and **42,217** bounding boxes annotated by domain experts. We have benchmarked 14 state-of-the-art algorithms on our MarineMaid dataset to reveal the strengths and limitations of existing general-purpose and domain-specific algorithms. The hierarchical and comprehensive experimental results provide valuable insights on how to develop practical and efficient marine visual perception algorithms to satisfy the domain requirements. To foster the further development of this direction, we will release our MarineMaid dataset with the acceptance of this paper.

## 1 INTRODUCTION

The unbounded depths of the ocean (Epstein et al., 1993; Ormond et al., 1997), rich with mysteries, have driven researchers to explore relentlessly, aiming to uncover its hidden secrets and valuable treasures (Thorne-Miller, 1999). The marine ecosystem (Epstein et al., 1993; Halpern et al., 2008)

is the most productive of all underwater ecosystems and shares immense ecological, social, and economic value. Performing marine study plays a significant role in protecting the marine environment and understanding marine science. However, marine research is limited compared with its volume. The ocean is continuously being polluted, leading to the migration of marine organisms (Perry et al., 2005) and species changes (Hiddink & Ter Hofstede, 2008; Poloczanska et al., 2013). Automatic marine life detection algorithms based on computer vision techniques are keenly required. Existing methods for monitoring and assessing marine ecosystem changes suffer from inefficiency and high labor costs. Nevertheless, utilizing computer vision techniques and deep learning algorithms can rapidly analyze marine images and videos, identifying species (Khan et al., 2023b) and tracking their migrations (Danovaro et al., 2010).

Object detection (Redmon et al., 2016; Ren et al., 2015; 2016; Liu et al., 2016), as a fundamental task, is to localize the interests of objects while discriminating the category information. The detected objects with bounding box annotations are important for species identification (Khan et al., 2023b), object tracking (Alawode et al., 2022), and object counting (Sun et al., 2023). To boost efficient marine object detection, there are several efforts proposed to build datasets (Zhuang et al., 2020; Liu et al., 2021b) and benchmarks (urp, 2020) to optimize powerful object detection algorithms. However, the categories (*e.g*, sea urchin, shark and *etc.*) are very limited, which are far away from satisfying to monitor a large range of marine creatures. Furthermore, the category information is not sufficient to satisfy the monitoring and surveying requirement, where biological traits (Miatta et al., 2021; Costello et al., 2015) are usually required. Furthermore, due to the essential monitoring purpose, the algorithms should also be able to detect a wide spectrum of objects (Zheng et al., 2023) and demonstrate strong generalization ability to unseen marine objects.

Vision-language models (VLMs) (Liu et al., 2024; 202, 2023; Team et al., 2023) achieve remarkable success thanks to large-scale datasets (Krishna et al., 2017; Gurari et al., 2018; Kazemzadeh et al., 2014; Shao et al., 2019). The VLMs could yield redundant visual descriptions based on the visual inputs, describing the objects with detailed attributes (*e.g.,* color, pose, activity, and *etc*). Despite the remarkable success of VLMs in a large number of visual understanding tasks, they are still poorly known to generate reasonable and domain-specific visual understanding for marine creatures. There are two main limitations when directly utilizing existing VLMs for marine visual understanding: data distribution shift and the lack of ability to localize and then describe the biological traits of the marine instances. The existing VLMs were mainly driven by datasets with dominant in-air objects and very limited marine objects, leading to unsatisfactory marine object understanding ability. There is a gap in the development and evaluation of VLMs for marine visual understanding for scientific research purposes. Furthermore, VLMs are optimized by redundant image-text pairs that succeed in holistic view understanding but struggle with detecting and understanding specific marine creatures with irregular boundaries/poses and also the ability to camouflage themselves into the background. Besides, generating the biological traits for detected instances with detailed descriptions of the spatial information/relationship between objects is also important to yield a complete analysis report. There is still a gap in utilizing existing algorithms and datasets for domain-specific marine research.

To fill this gap, we propose the first marine dataset and benchmark called **MarineMaid** to achieve robust and accurate marine visual understanding with detailed descriptions of biological traits from various aspects. Our dataset with rich biodiversity comprises 14,645 marine images with more than 42k human-labeled bounding boxes and instance captions, enhancing the understanding of the complex marine ecosystems. MarineMaid dataset enables various tasks, including open-vocabulary object detection, region-specific image/instance captioning and visual grounding specifically designed for marine creatures. Unlike existing image-text datasets with only short descriptions, MarineMaid provides a comprehensive and detailed description (average word length is 42) of the biological traits of the marine creatures from 4 aspects. To the best of our knowledge, our MarineMaid dataset is the first marine dataset to support marine monitoring and further analysis.

Based on our MarineMaid dataset, we have benchmarked the existing object detection, VLMs, and grounding algorithms to explore the boundary of these advanced algorithms to perform detailed marine visual understanding. Our MarineMaid stands as a novel and challenging testbed for both computer vision and marine research communities. Our main contributions are as follows:

- We propose the first region-level instance-caption pair dataset specifically designed for marine creatures, containing 12,873 fine-grained instance-captioning pairs and 42,217 BBOXs annotated by domain experts.

- We benchmark various marine visual understanding tasks including close-set object detection, open-vocabulary object detection, visual grounding, and instance captioning based on 14 state-of-the-art models.

## 2 RELATED WORK

**Existing Marine Research**. Unlike our everyday stuff, marine creatures usually possess significant diversity (a wide spectrum of poses, appearance, and patterns). Performing efficient marine visual understanding could harness the advanced algorithms (Li et al., 2021; Hong et al., 2020) to elevate marine research, conservation, and industrial endeavors. Existing marine datasets (*e.g.,* MAS3K (Li et al., 2020; 2021), WildFish (Zhuang et al., 2018), WildFish++ (Zhuang et al., 2020), SUIM (Islam et al., 2020)) have been proposed for promoting the recognition performance of marine organisms. However, most of these datasets only contain a few pre-defined categories without detailed captions, which limits the ability to accelerate the accumulation of detailed marine visual analysis through the creation of ocean databases and scientific data. Meanwhile, domain knowledge and expertise are required to do high-quality annotations (for both BBOX and caption), which is costly and time-consuming. In this work, we aim to propose a large-scale marine dataset with a wide spectrum of marine creatures.

**Object Detection**. Object detection is a fundamental computer vision problem (Lin et al., 2014; Ren et al., 2015; 2016), localizing the interests of objects and discriminating object categories simultaneously. The detection algorithms mainly fall into two categories: 1) one-stage algorithms (Liu et al., 2016; Ge et al., 2021; Redmon et al., 2016) perform localization and classification in parallel; 2) two-stage detection algorithms (Ren et al., 2015; 2016; He et al., 2017) generate the object proposals and then perform localization regression. However, these algorithms mainly perform close-set object detection. To address this limitation, open-vocabulary object detection (OVOD) (Zareian et al., 2021; Yao et al., 2023; Kim et al., 2023; Wang et al., 2023) aims to generalize beyond the limited number of pre-fixed classes during the training phase. The goal is to detect novel classes at the inference stage. The dominant way of performing OVD is to adopt a pre-trained visual encoder from a trained cross-modality alignment model, which is optimized by millions of image-text pairs from public websites. RegionCLIP (Kim et al., 2023) proposed to perform the regional visual feature and the textual conception alignment to promote the generalization ability to *unseen* categories.

**Vision-Language Understanding**. Vision-language models (VLMs) (Liu et al., 2024; 202, 2023; Team et al., 2023; Zhu et al., 2023; Liu et al., 2023a; Zheng et al., 2023; Li et al., 2022; 2023a) achieve remarkable success thanks to large-scale datasets such as Visual Genome (Krishna et al., 2017), VizWiz (Gurari et al., 2018), RefCOCO (Kazemzadeh et al., 2014), and Objects365 (Shao et al., 2019). VLMs bridge vision modality and text modality together to harness the power of large language models (LLMs) (OpenAI, 2022; 2023) and vision encoders (Dosovitskiy et al., 2020). Optimized by millions of image-text pairs, CLIP (Radford et al., 2021) demonstrated a strong zero-shot recognition ability for diverse images. BLIP (Li et al., 2022; 2023a) bootstraps vision-language pre-training from frozen pre-trained image encoders and frozen language decoders. However, these datasets only contain in-air objects or very limited marine objects, which is due to the poor ability of marine domain tasks. Furthermore, VLMs also struggle with the limited ability to perform region-level instance understanding following the user instructions.

## 3 DATASET AND APPROACH

**Overview**. We start by elaborating on the detailed dataset construction procedure of our MarineMaid and outlining the characteristics of our dataset, along with relevant statistics and explanations. We then provide the hierarchical and extensive experiments to benchmark marine object detection (including both close-set and open-vocabulary formulations), visual grounding, and instance captioning, revealing the strengths and limitations of existing algorithms.

### 3.1 DATASET CONSTRUCTION

**Data collection**. We collect images from the Internet. To maintain data quality and diversity, we manually reviewed all the images and removed duplicates or instances that did not align with the pre-defined categories. Existing datasets (Schuhmann et al., 2021) mainly utilized alt-texts to formulate the image-text pairs (*image-level*). However, the texts suffer from limited information (short captions), misalignment with the visual contents, and deviation from domain-specific requirements. In contrast, we generate comprehensive and contextually relevant *instance* captions based on the domain experts.

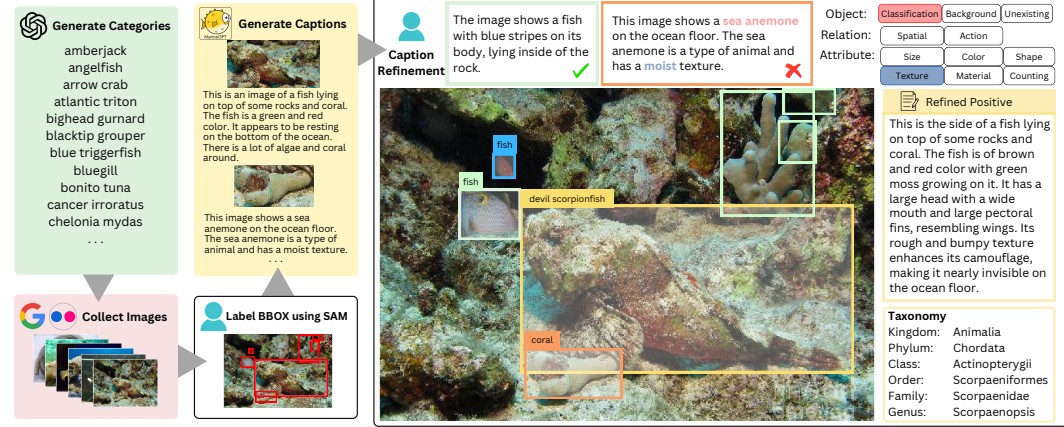

Figure 2: Overview of the dataset construction and data labeling pipeline, which can be summarized into five stages: 1) marine object categories are generated based on ChatGPT-3.5/GPT-4; 2) crawling corresponding marine images from the Internet (mainly from Google image engine and Flickr); 3) we employ SAM (Kirillov et al., 2023) model to label all marine objects present in each image by receiving the human prompts to iteratively obtain high-quality BBOX annotations; 4) the cropped image region based on the annotated BBOX from the whole image serves as the input to domain-specific VLM (MarineGPT (Zheng et al., 2023) used in this work) to generate object instance caption candidates for further human refinement; 5) domain experts refine the generated captions from some pre-defined aspects as the positive instance captions. We also provide the additional binary annotations from 11 diverse properties to identify some common prediction errors produced by VLMs and we regard these captions with binary annotations as negative captions (discussed in Supplementary material).

To promote labeling efficiency, we first utilize the marine-specific VLM MarineGPT to generate the candidate captions and the domain experts perform the refinement and revision from pre-defined aspects.

**Specific features**: 1) **Wide spectrum of marine object categories** (670 categories), varying from Cephalopods, Crustaceans, Sharks, Rays, Reptiles, Mammals, Aves, Corals, to Invertebrates. 2) **Hierarchical taxonomy**: including 6 coarse-to-fine granularities (Kingdom, Phylum, Class, Order, Family, Genus) by automatically querying the official Worms (Ahyong et al., 2024) database. 3) **Image diversity**: images were captured in various environmental conditions (*e.g.,* deep-sea, blurs, clutters, aquariums, markets, *etc*). Meanwhile, the images describe object instances from different aspects: activity events (*e.g.,*, hunting, reproductive, interactive, *etc*), life stages (*e.g.,* juvenile and adult), and image styles.

**Positive vs. Negative**. We also provide the *additional information* for the negatives to reveal common mistakes made by the models. We define 11 properties: classification, background, unexisting, spatial, action, size, color, shape, texture, material, and counting to summarize wrong captions. These negatives from VLMs and human post-processing offer more valuable insights compared to (Zhao et al., 2022; Yuksekgonul et al., 2022) that replaced correct objects with random noun phrases. These negative samples serve as the hard negatives to force the model to learn and recognize subtle feature differences.

**Data statistics**. Our dataset consists of 14,645 images, from a total of 670 categories. We manually labeled all identifiable marine life object, resulting in a total of 42,217 labeled bounding boxes. Among these, there are 24,197 large, 10,555 medium, and 7,465 small bounding boxes. There are 12,873 captions that have been refined by domain experts specializing in marine specialties, resulting in a superior level of quality. The average length of these refined positive captions is 42. Totally we have 22,321 refined and generated positive captions, and 12,431 generated negative captions.

## 3.2 LABELING PIPELINE

Our labeling pipeline encompasses three main stages: 1) BBOX labeling and refinement; 2) caption generation from VLM and refinement based on domain experts; and 3) cross-checking verification. **BBOX generation**. We first manually label bounding boxes for all the recognizable marine organisms

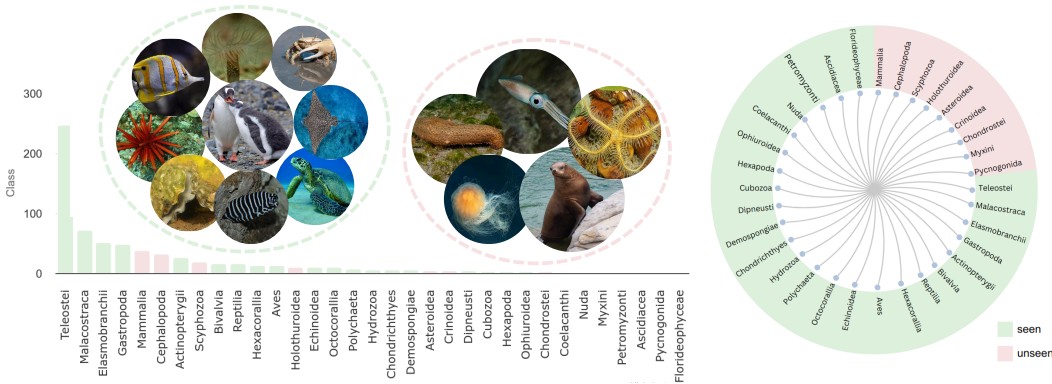

Figure 3: We provide the data statistics and the class distribution of MarineMaid at the Class-level granularity. The *seen* and *unseen* classes are split to perform open-vocabulary object detection.

within the image to perform dense labeling. To ensure the quality of the labeled BBOX annotations, we perform further refinement to ensure the whole instance (*e.g.,*, the transparent tail of the fish, and the slender legs of the shrimp) is accurately labeled. **Caption generation and refinement**. We first crop the marine object instance based on the BBOX annotations and feed the cropped image region to MarineGPT to generate the caption candidate based on the prompt "*describe this image in detail*". Please note that we only generate descriptive and informative captions based on image regions larger than 1024 pixels. Then based on the caption candidate, the domain experts do the refinement from four aspects: features (*e.g.,* unique characteristics, injuries, color, shape, size, *etc*), spatial information (*e.g.,* absolute and relative position), background and activity events (*e.g.,* individual or mutual). For each image, we only select one to perform caption refinement and we provide additional tags from the pre-defined 11 properties for the incorrect captions to formulate the hard negatives. **Cross-checking validation**. Finally, we perform the cross-validation based on two annotators to revise potential errors. Filtering: filters out some model-generated prompts or "unrecognizable" content. The annotators will cross-check and correct evident errors in captions, tags, and bounding boxes. Experts finally conduct inspections and verifications on uncertain objects to ensure accuracy and reliability. The construction of our MarineMaid dataset involves 16 domain experts with 624 human hours in total.

### 3.3 COMPARISON WITH EXISTING DATASETS AND BENCHMARKS

We provide a direct comparison with existing general-purpose and domain-specific datasets in Table 1 from various aspects: the data/annotation volume; annotation type; whether the image/instance captions provided and the average word length of these corresponding captions; and Taxonomy for hierarchical classification and understanding. MarineMaid dataset possesses three main advantages over existing datasets: 1) compared with existing marine datasets, which mainly provide the BBOX and mask annotations, the MarineMaid dataset provides detailed instance captions for the object instance besides the BBOX annotations. 2) Compared with general-purpose datasets that contain a large scale of image/instance captions, our provided instance captions are significantly longer (42 vs. 12), describing diverse biological traits of marine creatures. 3) Compared with the existing Wildfish++ dataset with both taxonomy and visual descriptions from the domain experts, MarineMaid is 10 times larger and contains a wide range of marine creatures while wildfish++ only focuses on fish.

## 4 EXPERIMENTS

In this section, to comprehensively evaluate the effectiveness of marine visual understanding, we choose three representative visual understanding tasks, including object detection (both close-set and open-vocabulary settings), region-level instance captioning, and visual grounding. We benchmark the existing state-of-the-art algorithms for corresponding tasks on our MarineMaid dataset.

### 4.1 OBJECT DETECTION

**Experimental settings**. **Dataset split**. We construct *seen/unseen* split following three settings: 1) Class-level: We consolidated the 670 categories into 33 categories based on their taxonomic Class (Ahyong et al., 2024). Certain object categories (*e.g.,*, bryozoa), are classified at a higher level of granularity (Phylum) and are therefore excluded from the Class-level categories. As illustrated in Fig. 3, we adopt 24 Classes as *seen* and the other 9 Classes as *unseen*. 2) Intra-Class: Intra-Class

Table 1: We provide a direct comparison between our MarineMaid dataset with both general-purpose datasets and marine-specific datasets. – indicates that the numbers were either not reported in their publications or we are unable to conduct statistical analysis.

| Datasets | Categories | Images | Annotations | Image/Instance Captions | Avg. Length | Taxonomy |
|---|---|---|---|---|---|---|
| DUO (Liu et al., 2021a) | 4 | 7,782 | $74,515_{bbox}$ | None | None | × |
| SUIM (Islam et al., 2020) | 8 | 1,525 | $1,525_{mask}$ | None | None | × |
| MAS3K (Li et al., 2020) | 37 | 3,103 | $3,103_{mask}$ | None | None | × |
| UIIS (Lian et al., 2023) | 7 | 4,628 | $4,628_{mask}$ | None | None | × |
| SEAMPD21 (Boulais et al., 2021) | 130 | 28,328 | $90,000_{bbox}$ | None | None | × |
| Wildfish (Zhuang et al., 2018) | 1,000 | 54,459 | $54,459_{cls}$ | None | None | × |
| FishNet (Khan et al., 2023a) | 17,357 | 94,532 | $114,375_{bbox}$ | None | None | ✓ |
| Wildfish++ (Zhuang et al., 2020) | 2,348 | 103,034 | $103,034_{cls}$ | 3,187 | 56 | ✓ |
| nocaps (Agrawal et al., 2019) | – | 15,100 | Caption | 166,100 | – | × |
| Redcaps (Desai et al., 2021) | – | 12,011,121 | Caption | 12,011,121 | 9 | × |
| Pascal Sentences (Rashtchian et al., 2010) | 20 | 1,000 | Caption | 4,998 | 10 | × |
| SBU Captions (Ordonez et al., 2011) | 81 | 1,000,000 | Caption | 1,000,000 | 12 | × |
| MarineMaid | 670 | 14,645 | $42,217_{bbox}$ | 12,873 (Refined) / 34,752 (All) | 42 / 33 | ✓ |

categorization is obtained by retrieving object categories at the Class-level. Under this setting, we have 555 *seen* categories and 109 *unseen* categories. 3) Inter-Class: we choose 1 object category from every 4 object categories in each Class as the *unseen* and the other 3 object categories as *seen*. We omit the Class that contains less than 4 object categories. With this setup, there are 482 *seen* categories and 161 *unseen* categories.

**Implementation details. Close-set object detection setting**. We mainly include 3 close-set object detection algorithms (Faster-RCNN (Ren et al., 2015), GridRCNN (Lu et al., 2019) and YOLOX (Ge et al., 2021)) and report the $mAP_{50}$ of 24 *seen* categories under three settings(Class-level, Intra-Class and Inter-Class). Our implementation of these models is based on MMDetection (Chen et al., 2019) using the official experimental setting. Please note that we do not evaluate these close-set object detection algorithms on the *unseen* categories. **Open-Vocabulary Object Detection** We evaluate the performance of 3 open-vocabulary object detection algorithms (RegionCLIP (Zhong et al., 2022), UniDetector (Wang et al., 2023) and DECOLA (Cho & Krähenbühl, 2023)) on our MarineMaid dataset. For RegionCLIP (Zhong et al., 2022), we follow the official experimental setting and fine-tune the model on our MarineMaid dataset. We adopt the single-dataset training strategy for UniDetector (Wang et al., 2023) to continuously optimize it in an end-to-end fashion. For DECOLA (Cho & Krähenbühl, 2023), we utilize their best-performing model with Swin-B backbone (phase 1) as the pre-trained model. We inherit the language-conditioned detection training procedure of DECOLA while keeping other configurations the same. At the evaluation stage, we report the quantitative results for both *seen* and *unseen* categories. The $mAP_{50}$ is computed to comprehensively evaluate the ability of models to detect overall marine object instances.

**Comparison and analysis.** We report the quantitative result in Table 2 and all the experiments are conducted following the same train/val data split. We have observed that the existing generalist object detection algorithms still face challenges when optimized by underwater images in providing accurate object localization. This can be attributed to two potential reasons: 1) the huge conception distance between the in-air object categories and marine object categories; and 2) the diversity of underwater data and the inherent challenges of underwater scenes make it difficult to extract features. Furthermore, as demonstrated, open-vocabulary detection algorithms, continuously fine-tuned on the MarineMaid dataset, typically exhibit improved detection performance even on *seen* categories compared to close-set counterparts. We attribute such promoted performance to the optimization through large-scale datasets with redundant supervised training data during the pre-training procedure. We present a qualitative comparison of the results in Fig. 4 under Class-level setting. DECOLA exhibits superior performance in semantic and object localization when detecting *seen* objects. However, when it comes to *unseen* objects, the models struggle to accurately classify the object category. In both Intra-Class and Inter-Class settings, DECOLA is the sole model to gain an advantage over the fine-grained marine species. We attribute such powerful fine-grained recognition ability to its language-conditioned query selection strategy.

## 4.2 INSTANCE CAPTIONING

**Experimental settings**. We benchmark off-the-shelf VLMs from two aspects: *image-level* and *region-level*. The former image-level VLMs (LLAVA (Liu et al., 2024), MiniGPT-4 (Zhu et al.,

Figure 4: The qualitative comparison between different algorithms under the Class-level setting. *The left part of the dashed line*: the results of *seen* category. *The right part*: the results of *unseen* category.

Table 2: Quantitative object detection (close-set and open-vocabulary) results on our MarineMaid dataset. − indicates the results cannot be computed under the settings.

| Method | Seen | | | Unseen | | |
|---|---|---|---|---|---|---|
| | Class-level | Intra-Class | Inter-Class | Class-level | Intra-Class | Inter-Class |
| FasterRCNN (Ren et al., 2015) | 28.7 | 17.6 | 16.7 | - | - | - |
| YOLOX (Ge et al., 2021) | 27.5 | 21.7 | 21.0 | - | - | - |
| GridRCNN (Lu et al., 2019) | 32.7 | 28.1 | 28.6 | - | - | - |
| UniDetector (Wang et al., 2023) | 31.5 | 23.3 | 24.1 | 8.2 | 0.4 | 0.7 |
| RegionCLIP (Zhong et al., 2022) | 39.8 | 34.1 | 29.8 | 12.2 | 6.2 | 0.4 |
| DECOLA (Cho & Krähenbühl, 2023) | **66.7** | **88.8** | **86.9** | **37.7** | **51.6** | **52.3** |

2023), BLIP2 (Li et al., 2023b) and InstructBLIP (Dai et al., 2024)) were optimized by image-level captions and lacked the ability to understand specific object instances. We evaluate these image-level VLMs based on the following user instruction: "*describe the object in this figure*". The latter region-level VLMs (GroundingLMM (Rasheed et al., 2023), GPT4RoI (Zhang et al., 2023)) were optimized by paired image region prompts and the corresponding instance captions. We provide the BBOX annotation in the given text prompt following the experimental setting of (Rasheed et al., 2023; Zhang et al., 2023). We perform the evaluations based on the positive instance captions to analyze their capability in describing marine instance objects. To quantitatively measure the performance of various algorithms, we adopt the widely used captioning metrics (Hessel et al., 2021; Vedantam et al., 2015; Banerjee & Lavie, 2005; Lin, 2004; Papineni et al., 2002) (including CLIPScore, RefCLIPScore (Hessel et al., 2021), CIDEr (Vedantam et al., 2015), BLUE-4 (Papineni et al., 2002), METEOR (Banerjee & Lavie, 2005) and Rouge (Lin, 2004)) to compute quantitative results in Table 3. Besides the human-constructed instance captions proposed in our MarineMaid dataset, we also construct a starting sentence to include the category information for the selected object instance: "This is a <Category Name>.", where the <Category Name> is the placeholder to compensate the scientific category-level information of each instance. In this way, by penalizing generated plausible but not domain-specific responses (*e.g.,* "fish", "animal" and "mammal"), we encourage the model to generate the scientific captions to satisfy the domain requirements.

**Implementation details**. We perform the evaluation only based on the released official models provided by various algorithms on our MarineMaid dataset and our experiments were conducted using an NVIDIA L20 GPU. For LLAVA (Liu et al., 2024), we choose its V1.5-7b version for evaluation.

Table 3: Results of various algorithms (image-level and region-level) on instance captioning.

| Method | CLIPScore↑ | RefCLIPScore↑ | CIDEr↑ | BLUE-4↑ | METEOR↑ | Rouge↑ |
|---|---|---|---|---|---|---|
| LLAVA (Liu et al., 2024) | 73.78 | 72.27 | 4.93 | 8.77 | 7.70 | 20.76 |
| MiniGPT-4 (Zhu et al., 2023) | 74.48 | 73.43 | 5.72 | 7.18 | **16.90** | **28.03** |
| BLIP2 (Li et al., 2023b) | 76.22 | 73.73 | 9.96 | 8.16 | 5.95 | 18.96 |
| InstructBLIP (Dai et al., 2024) | **76.60** | **75.25** | **12.09** | **13.94** | 7.40 | 21.31 |
| GroundingLMM (RefCOCOg) (Rasheed et al., 2023) | 73.04 | 70.97 | 4.37 | 4.39 | 4.60 | 16.37 |
| GroundingLMM (VG) (Rasheed et al., 2023) | 71.15 | 69.04 | 4.06 | 2.47 | 4.11 | 15.22 |
| GPT4RoI (Zhang et al., 2023) | 71.28 | 68.71 | 3.53 | 2.81 | 4.07 | 15.08 |

| | | | |
|---|---|---|---|
| LLAVA | A black fish with a white stripe on its tail swims through the water. | A person is holding a fish in their hand, showing off its shiny scales. | A small crab is sitting inside a pink shell, possibly a sea anemone. |
| MiniGPT-4 | this is a black fish with white stripes swimming in a tank filled with water. the fish has a long, slender body and a small mouth with sharp teeth. it is swimming towards the camera, its fins flapping rapidly as it moves through the water. the background of the image is a dark, rocky cave with a small amount of algae growing on the walls. | this image shows a group of people holding a large fish on a boat. the fish is a blue marlin, which is a large and powerful fish that is prized for its meat and sport fishing. the people in the image are wearing fishing gear and appear to be preparing to release the fish back into the water. | this image shows a large, pink sea anemone with its tentacles spread out, surrounded by other sea creatures such as a crab and a fish. the anemone appears to be in a coral reef environment, with other coral and seaweed visible in the background. the image is taken underwater, with the sunlight filtering through the water to create a beautiful, colorful scene. |
| BLIP2 | a black slender ray swimming in an aquarium | a tuna caught on a fly fishing trip in the bahamas | a close up of the inside of a sea urchin |
| Instruct BLIP | a black and white fish swimming in an aquarium | a man holding a large fish on a boat | a close up of a large pink shell on the ocean floor |
| Grounding LMM | a black fish with a white stripe on its head | a large fish being held by two men | a purple and pink sea star |
| GPT4RoI | a black fish swimming in the water | a man holding a fish | a purple and pink organism |
| GT | This is a black ghost knifefish swimming in the water. The water is clear and there is a piece of rock, aquatic plants and gravel in the background. The body of the fish is black and it has a long tail with black and white stripes. | The object in the middle is a tuna being held in the left hand of a fisherman in a boat. The fish is silver with purple and green iridescent markings on its body. Its dorsal fins are separate and both steep and pointed. There is another person standing to the fisherman's left. | This is a close-up of a pink barrel sponge seen from above the animal. The inside surface is smooth and layered and the outside surface appears spikey and rough. The appears to be a smaller blue sponge behind the object. |

Figure 5: The qualitative results of different algorithms on marine object instance understanding. Best viewed in color.

The language model of MiniGPT-4 (Zhu et al., 2023) is set to LLaMA-2 (Touvron et al., 2023). As for the GroundingLMM (Rasheed et al., 2023), we report the results of the models fine-tuned on RefCOCOg dataset (Kazemzadeh et al., 2014) and Visual Genome (VG) dataset (Krishna et al., 2017), respectively.

**Comparison and analysis**. All the quantitative results are reported in Table 3. Please note that CLIPScore and RefCLIPScore (Hessel et al., 2021) are computed based on the whole image. We observe that image-level VLMs achieve various scores when there are human-constructed reference captions. LLAVA (Liu et al., 2024) and BLIP2 (Li et al., 2023b) achieve very poor outputs on CIDEr (Vedantam et al., 2015) and BLUE-4 (Papineni et al., 2002) since these two tend to generate very short answers and they also make some wrong recognitions. InstructBLIP (Dai et al., 2024) performs best on CLIPScore, RefCLIPScore (Hessel et al., 2021), CIDEr (Vedantam et al., 2015) and BLUE-4 (Papineni et al., 2002). This indicates that the instruction-following tuning could heavily promote the ability of the models to understand the instances following the user instructions. However, the generated instance captions are still too short to satisfy the domain requirement. Region-level VLMs also achieve very poor results since they cannot accurately localize the specific marine instances by the user-provided BBOX prompts. Thus, the region-level VLMs still describe the whole image and yield wrong captions as demonstrated in Fig. 5. There is still a gap when utilizing existing VLMs for marine instance understanding.

## 4.3 GROUNDING

**Experimental settings.** We finally demonstrate the performance of existing general-purpose grounding algorithms in handling marine visual localization. Using our captions as prompts, we apply these algorithms to generate target bounding boxes, which are then compared to the ground truth bounding boxes in our dataset. Specifically, we examine GroundingDINO (Liu et al., 2023b) and GroundVlP (Shen et al., 2023). These models are not fine-tuned on our training set and are directly evaluated on our validation set. Additionally, captions that are negatives, empty, and with no noun phrases detected by nltk package are excluded to ensure a smooth evaluation process. The results are reported in Table 4, following the default evaluation metrics (Recall used in GroundingDINO (Liu et al., 2023b) and accuracy for GroundVLP (Shen et al., 2023)). To guarantee a proper configuration of the evaluation environment settings, we meticulously adhered to the instructions and evaluation program provided by the authors (discussed in Supplementary).

(a)Query: This is a bubble-tip anemone under aquarium lighting. It has bulbous tops on its tentacles that are closely packed together in purple color, with a large protruding spot at the tips.

(b)Query: This is an angel shark swimming on the ocean floor. It is large in size with a flat body. Its tail is long with two raised dorsal fins. The color of its body is similar to the ocean floor.

(c)Query: This is a shark in a tank. It appears to be swimming in the water.

(d)Query: This is a sea urchin with long, sharp black spines on its body. It is located in a body of water with other sea creatures nearby. There is a purple and yellow fish on top of it.

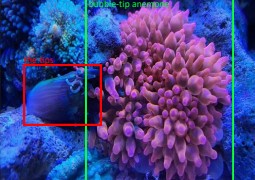 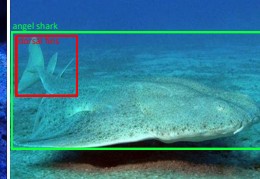 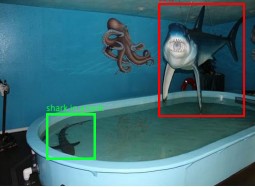 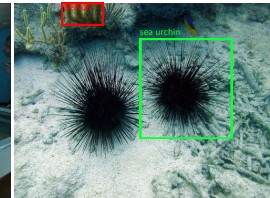

Figure 6: Results of GroundingDINO (Liu et al., 2023b) (a,b) and GroundVLP (Shen et al., 2023) (c,d) on our MarineMaid dataset. Green texts and BBOXs indicate the query and GT respectively. Red texts and BBOXs indicate model-generated predictions and corresponding BBOX outputs.

Table 4: Quantitative visual grounding results on MarineMaid dataset.

| Method | Evaluation Metric | Seen | | | Unseen | | |
|---|---|---|---|---|---|---|---|
| | | Class-level | Intra-Class | Inter-Class | Class-level | Intra-Class | Inter-Class |
| GroundingDINO (Liu et al., 2023b) | R@1 | 38.8 | 37.5 | 37.2 | 46.8 | 46.8 | 45.6 |
| GroundingDINO (Liu et al., 2023b) | R@5 | 67.3 | 65.5 | 66.2 | 76.9 | 76.8 | 74.5 |
| GroundingDINO (Liu et al., 2023b) | R@10 | 78.0 | 76.0 | 76.8 | 84.9 | 84.9 | 83.3 |
| GroundingVLP (Shen et al., 2023) | Accuracy | 19.8 | 45.8 | 42.8 | 34.3 | 35.6 | 52.4 |

**Implementation details.** For GroundingDINO (Liu et al., 2023b), we employ the best Swin-B pre-trained model (MM-GDINO-B*) as the backbone. The evaluation is conducted on a single GeForce RTX 2080 Ti. Other configurations are consistent with the original paper. For GroundVLP (Shen et al., 2023), we use the ALBEF and Swin-B Detic (Zhou et al., 2022) models provided by the authors, evaluated on our validation dataset.

**Comparison and analysis.** As reported in Table 4, there is an obvious performance drop (still unsatisfactory performance) when utilizing the two grounding algorithms on marine creature local-ization. We attribute this to the gap in knowledge between everyday objects and marine creatures. Furthermore, as depicted in Fig. 6, these algorithms struggle to accurately recognize and locate target marine creatures, being hindered by the knowledge acquired from in-air objects. For instance, in Fig. 6 (c), GroundVLP (Shen et al., 2023) mistakenly identifies a shark as a cow and fails to locate the correct target based on the described action in the caption. Conversely, GroundDINO (Liu et al., 2023b) (b) correctly identifies the angle shark, but mistakenly recognizes its tail fin as dorsal fins, further revealing the lack of ability to perform accurate marine visual grounding.

## 5 DISCUSSIONS AND CONCLUSION

**New benchmark.** The proposed MarineMaid serves as a novel comprehensive and diverse benchmark meticulously curated for marine research. Our dataset is introduced to enhance the assessment of existing algorithms for marine visual understanding. It includes a wide range of marine creatures across various environments, providing a valuable benchmark for testing and developing new models.

**Broader impact.** The study of marine creatures has several important applications, such as identifying and safeguarding rare animal species, preventing wildlife trafficking, and aiding in search-and-rescue operations. Our dataset deliberately excludes any military or sensitive scenes, ensuring its focus remains on benign and beneficial applications.

**Limitation.** Even though we tried our best to cover the most common marine creatures, we have to admit that the amounts of existing marine creatures are much larger than the included marine object categories. Our dataset will be continuously growing to include more marine object categories.

**Conclusion.** In this work, we propose the first large-scale marine datasets to enable both object detection and vision-language understanding. Our dataset supports various tasks, including *close-set object detection*, *open-vocabulary object detection*, *instance captioning*, and *grounding*. The comprehensive evaluation sheds light on the strengths and limitations of both general-purpose and domain-specific algorithms.

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

## A  APPENDIX

### A.1  DATA ANNOTATION

**BBOX annotation**. We develop a bounding box annotation platform as shown in Fig. 7. The left side represents the image with the point prompts from the users. We embed the segment anything model (SAM) as our labeling engine. The green dots indicate selected areas and the red dots indicate unselected areas. The right slide is the labeled bounding box and corresponding instance mask output automatically generated by SAM.

**Captions refinement**. We develop a user-friendly caption annotation platform as shown in Fig. 8. On the left is the image and the object bounding boxes, and on the right is the intention description of the object. We divided the images into 100 per subset and assigned them to experts so that the duration of each consecutive work is not too long to ensure the quality of the annotation. We select one salient object to perform caption refinement and tag "*Refined Positive*" as shown in Fig. 8. The generated captions are then distinguished into "*Generated Positive*" and "*Generated Negative*" for the remaining objects. Fig. 9 is the example of "*Generated Negative*". We provide additional tags from three aspects (Object, Relation, and Attribute) including 11 properties, Table 5 presents the example and statistic.

Table 5: The detailed explanations of the constructed 11 attributes and corresponding data statistics for the generated negative captions.

| | Properties | Example | Number |
|---|---|---|---|
| Object | Classification | This is a yellow fish. *vs.* This is a yellow coral. | 6,875 |
| | Background | The turtle is in the ocean. *vs.* The turtle is in the sky. | 1,343 |
| | Unexisting | The shark has a long tail. (there is no tail in the image) | 3,264 |
| Relation | Spatial | This fish is under the coral. *vs.* This fish is on the coral. | 816 |
| | Action | The penguin is walking. *vs.* The penguin is sitting. | 938 |
| Attribute | Size | The shark is large. *vs.* The shark is small. | 271 |
| | Color | This is a yellow fish. *vs.* This is a blue fish. | 2,031 |
| | Shape | This is a oval seashell. *vs.* This is a triangle seashell. | 312 |
| | Texture | The seashell is smooth. *vs.* The seashell is rough. | 321 |
| | Material | The fish is probably made of plastic. | 316 |
| | Counting | There are three penguins *vs.* There are four penguins. | 831 |

### A.2 EXAMPLES AND DATA DIVERSITY

**Examples**. We provide some image examples with the detailed bounding box instance caption annotations in Fig. 10. We encourage the readers to pay more attention to the generated instance captions. The instance captions describe the appearance of the object instance, action, event, the relationship between the selected instance with other instances, and more advanced biological traits.

**Diversity and data composition**. We provide the illustration about the data diversity of our constructed MarineMaid dataset in Fig. 11. We only provide some images from some categories for better illustration.

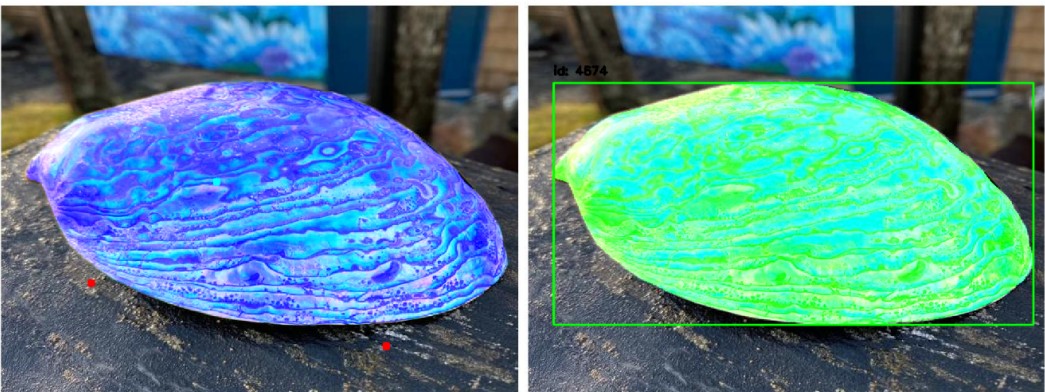

Figure 7: Screenshot of the BBOX annotation platform. *Left:* input point prompt. *Right:* the labeled instance BBOX and mask.

## B MORE EXPERIMENTS

### B.1 DATA SPLIT

We implement a consistent splitting strategy for each dataset: class-level, intra-class, and inter-class. For the training set, 80% of the images containing objects from seen categories are randomly sampled. The remaining 20% of seen objects, along with all unseen objects, are allocated to the validation set. To assess performance on both seen and unseen objects, the validation set is further divided into *val_seen* and *val_unseen* based on categories. Images containing both seen and unseen objects are manually reassigned to the validation set, resulting in duplicated images in both *val_seen* and *val_unseen*, each with different annotations. Statistics and details can be found in Table 6.

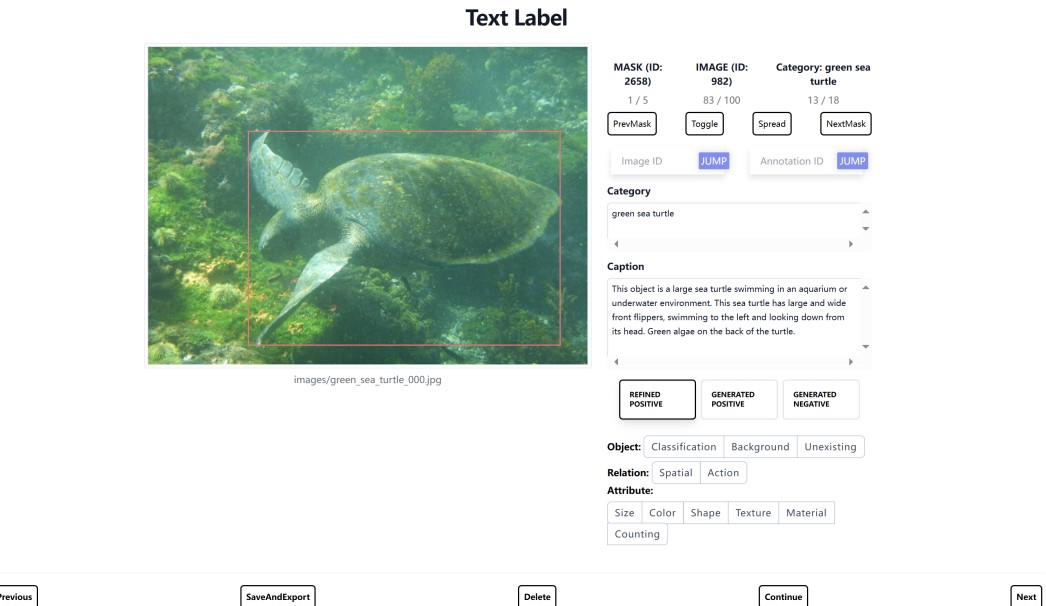

Figure 8: Screenshot of our developed caption refinement platform for generating the "*Refined Positive*". The domain experts are required to modify and edit the accurate and detailed biological traits for the selected marine instance.

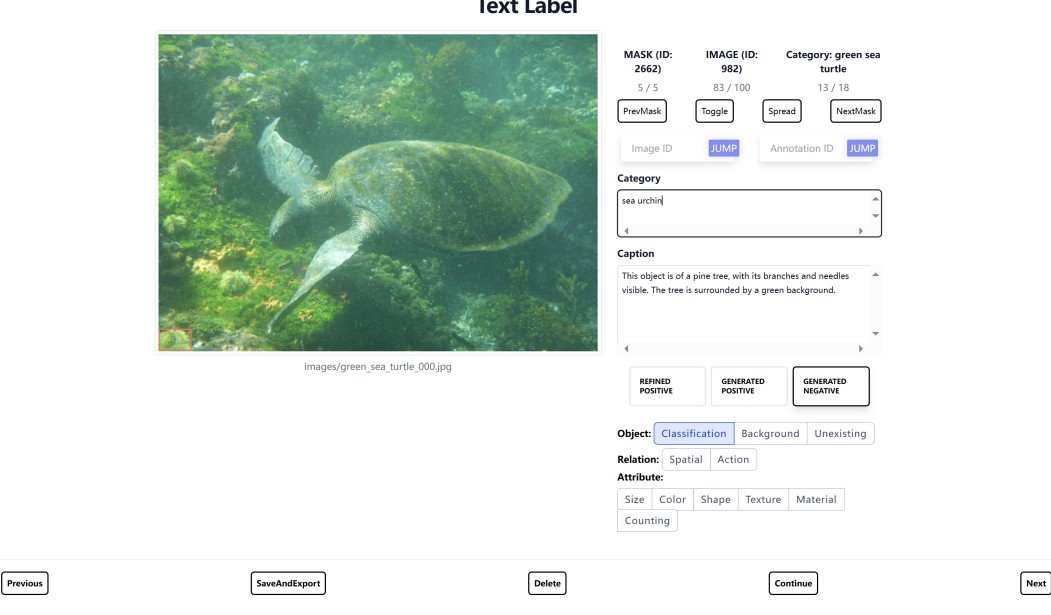

Figure 9: Screenshot of our developed caption refinement platform for generating the "*Refined Negative*". The annotators are asked to provide additional attribute annotations (wrong types) for the negative captions.

Table 6: Data split for performing the open-vocabulary object detection. We provide detailed data splitting under each setting.

| | Class-Level | | | | Intra-Class | | | | Inter-Class | | | |
|---|---|---|---|---|---|---|---|---|---|---|---|---|
| | Train | val | val (seen) | val (unseen) | Train | val | val (seen) | val (unseen) | Train | val | val (seen) | val (unseen) |
| # of Images | 9,291 | 5,298 | 2,743 | 2,963 | 9,312 | 5,301 | 2,746 | 2,963 | 8,163 | 6,062 | 2,943 | 4,023 |
| # of BBOX | 27,560 | 14,540 | 9,071 | 5,469 | 28,166 | 13,992 | 8,523 | 5,469 | 23,561 | 17,920 | 9,960 | 7,960 |
| # of Captions | 22,739 | 11,897 | 6,699 | 5,198 | 22,709 | 11,984 | 6,786 | 5,198 | 19,232 | 14,790 | 7,527 | 7,263 |

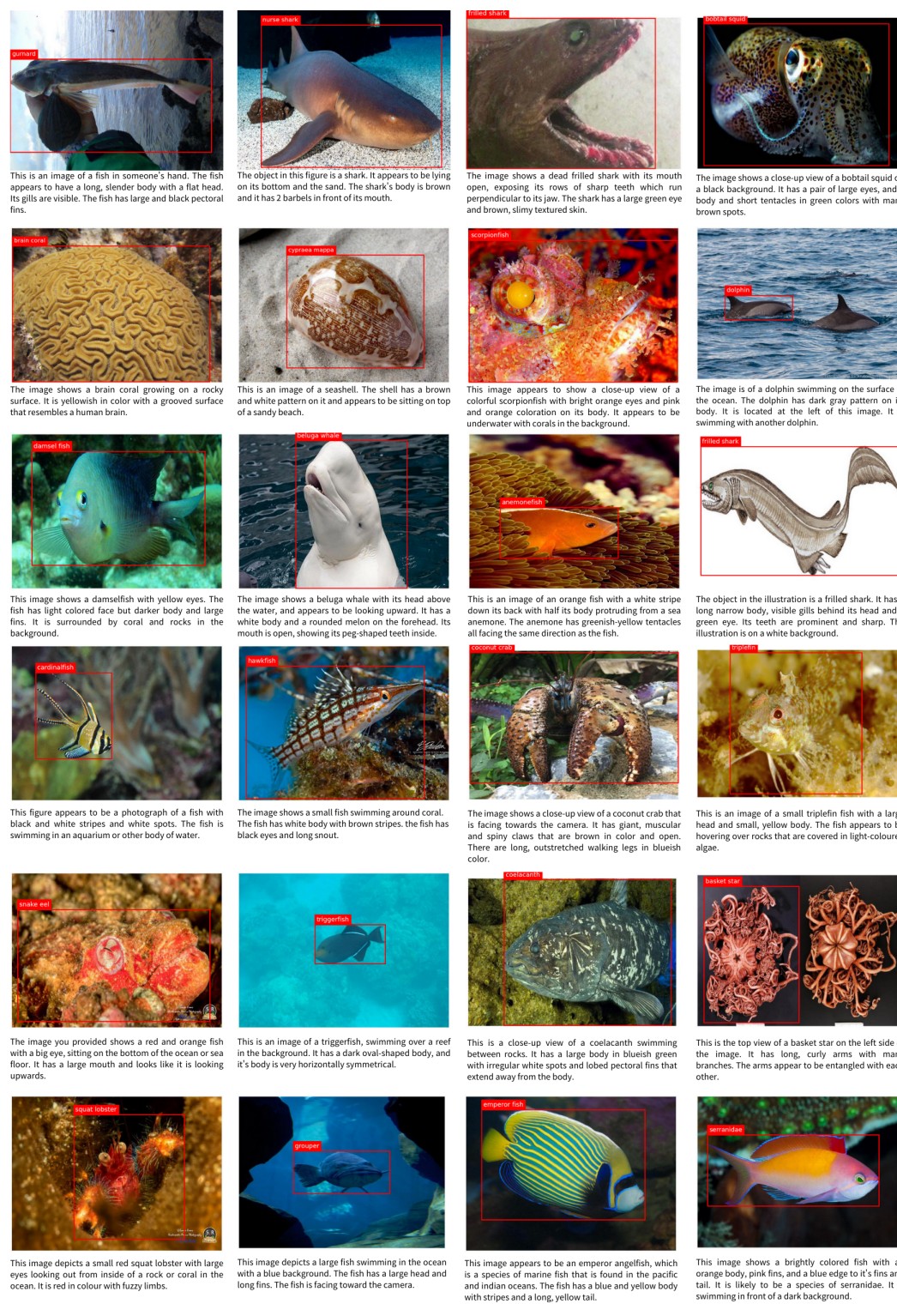

Figure 10: The example images with the bounding box annotations and instance captions from our MarineMaid dataset. Best viewed in color.

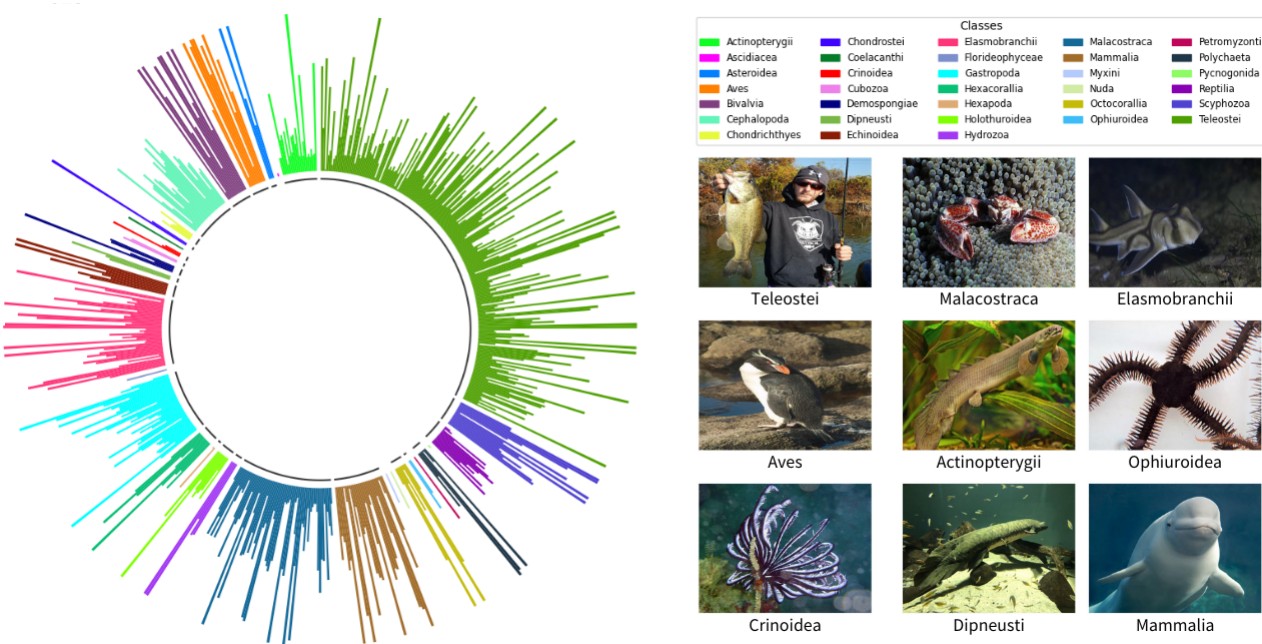

Figure 11: We present the the data distribution of our MarineMaid dataset at the "Class" level. We also provide images from some selected Classes for illustration.

Table 7: Results of MiniGPT-4 under two settings on our MarineMaid dataset.

| Method | CLIPScore↑ | RefCLIPScore↑ | CIDEr↑ | BLUE-4↑ | METEOR↑ | Rouge↑ |
|---|---|---|---|---|---|---|
| Vanilla | 74.48 | 73.43 | 5.72 | 7.18 | 16.90 | 28.03 |
| Fine-tuned | 77.96 | 77.51 | 17.36 | 14.79 | 16.90 | 33.71 |

## B.2 INSTANCE CAPTIONING

Due to the constraint of the computational power, we select the representative MiniGPT-4 (with LLaMA2 7B version) to do the fine-tuning on our MarineMaid dataset. Please note that the testing set is withheld for evaluation purposes. We finetuned the MiniGPT-4 on our dataset on 4 NVIDIA A100-40GB for 5 epochs and we set other training parameters to follow the same as its original paper. We report the experimental results under the two settings (vanilla and fine-tuned) in Table 7. We observe that further fine-tuning could help improve the instance understanding performance. But there is still large room for further improvement. The sole fine-tuning cannot fully solve our problem and domain-specific design and modifications are required.

## B.3 GROUNDING

Following a similar experimental setting, we select GroundindDINO to perform the grounding experiments. To guarantee the proper configuration of the evaluation environment settings, we meticulously adhered to the instructions and evaluation program provided by the original official implementations. We further fine-tune GroundingDINO on our training dataset to measure the performance. We use the same pre-trained model (MM-GDINO-B*) to optimize the model on our dataset. The results are reported in Table 8. We observe an observable improvement in R@1 across all validation settings, though there is a decline in performance in R@5 and R@10. This indicates that after fine-tuning, the model becomes more proficient at identifying the target object within the image based on the query but also detects additional regions that do not correspond to the ground truth bounding box. However, we also acknowledge the performance drop of the R@10. We attribute such performance drop to the specific feature of our constructed MarineMaid dataset (we only aim to ground one instance based on the captions).

Table 8: Performance comparison of GroundingDINO under two settings: Vanilla and Fine-tuned.

| | | Seen | | | Unseen | | |
|---|---|---|---|---|---|---|---|
| | | Class-Level | Inter-Class | Inter-Class | Class-Level | Inter-Class | Inter-Class |
| Vanilla | R@1 | 38.8 | 37.5 | 37.2 | 46.8 | 46.8 | 45.6 |
| | R@5 | 67.3 | 65.5 | 66.2 | 76.9 | 76.8 | 74.5 |
| | R@10 | 78.0 | 76.0 | 76.8 | 84.9 | 84.9 | 83.3 |
| Fine-tuned | R@1 | 65.7 | 62.1 | 64.5 | 71.6 | 70.6 | 70.3 |
| | R@5 | 71.9 | 66.1 | 63.5 | 76.0 | 73.6 | 71.4 |
| | R@10 | 74.2 | 67.7 | 64.1 | 78.5 | 74.8 | 71.7 |

