# OpenReview forum: "MarineMaid: Dataset and Benchmark on Detecting and Understanding Marine Creatures"
_ICLR.cc/2025/Conference — ICLR 2025 Conference Withdrawn Submission_

### Official Review · Reviewer_9chD · 2024-10-27

**Soundness:** 3
**Presentation:** 3
**Contribution:** 2
**Rating:** 5
**Confidence:** 4

**Summary:**

In this paper, the authors propose the MarineMaid dataset designed for marine visual understanding. Object detection and vision-language understanding are the fundamental tasks evaluated on the proposed dataset. The authors use 5 steps to annotate the dataset, including  Generating Categories, Collecting Images, Labelling BBOX using SAM, Generating Captions, and Refining Captions. In the experiments, the authors adopt popular detectors and instance captioning methods in the evaluation.

**Strengths:**

1. The paper is well wirtten and easy to follow.
2. The dataset contains more caption and taxonomy than other datasets.
3. Both close-set and open-vocabulary detections have been compared.

**Weaknesses:**

1. As shown in Table 1, MarineMaid dataset provides better captions and taxonomy, but has less instances than some other datasets. This is one of the weaknesses of the dataset.
2. The paper does not propose any baseline method for marine object detection and understanding. The contribution seems limited.
3. Only mAP50 is used in the evaluation. More metrics should be used for comparison.

**Questions:**

1. As shown in Table 2, the DECOLA method performs much better than other methods. I am curious why this happens. Do all the methods use the same training set? What are the pre-trained datasets used for each method? Is it a fair comparison?
2. As shown in Table 1, there are several large-scale datasets with more categories, images and box annotations. I am curious why do not extend these existing datasets with rich caption annotations instead of creating a new dataset with much fewer images.

---

### Official Review · Reviewer_NdNE · 2024-10-28

**Soundness:** 2
**Presentation:** 1
**Contribution:** 1
**Rating:** 1
**Confidence:** 5

**Summary:**

The authors propose a dataset for marine monitoring, focused on object detection, captioning and grounding tasks. The dataset is collected from internet and the annotation process is carried out by the authors. The authors then benchmark state-of-the-art algorithms on these tasks.

**Strengths:**

1) The dataset can be a useful resource for research in underwater computer vision.

**Weaknesses:**

1) While I understand that the authors may be non-native English speakers, the current state is, in my opinion, below the threshold of acceptability, from this perspective. For example, these are two sentences from the introducion:
- “average word length is 42”: the authors surely mean the sentence length, not the word length.
- “To the best of our knowledge, our MarineMaid dataset is the first marine dataset to support marine monitoring and further analysis”: this is also clearly false and inconsistent with the references provide by the authors in the related work section.

2) The labeling process is unclear. Fig. 2 says that “prompts” are provided to SAM for bounding box generation, but this is not mentioned in the text. While it is briefly mentioned in the appendix, I believe this should be in the main text, as it is the only kind of “methodological” contribution provided by this paper. The overall description of the labeling and revision pipeline is very vague and confusing.

3) The experimental section is straightforward and not very useful. It is not clear whether the quality of annotations in MarineMaid is better than other datasets, and especially how/if it impacts the performance on downstream tasks.

**Questions:**

The authors should clarify in more detail the labeling process. How many experts were involved? What kind of experts were they? How is the effort required from each expert?

---

### Official Review · Reviewer_CcBY · 2024-11-01

**Soundness:** 3
**Presentation:** 3
**Contribution:** 3
**Rating:** 6
**Confidence:** 4

**Summary:**

This paper introduces the MarineMaid dataset, tailored for marine visual understanding. The dataset includes high-quality bounding boxes for object detection and fine-grained instance-caption pairs for vision-language tasks. Additionally, the authors benchmark current algorithms on the proposed dataset.

**Strengths:**

+ The paper is easy to follow and clear.
+ The motivation for the dataset is sound, providing both detailed instance-level captions and object bounding boxes, which are valuable for future domain-specific tasks.
+ The labelling process uses auto-labelling models and also involves human data curation to make sure the quality.
+ The experiments are comprehensive, including benchmarking across different tasks and models.

**Weaknesses:**

- The bounding boxes appear to be primarily generated using SAM models. How accurate are these bounding boxes for small fish groups? Besides, when fish are challenging to detect, how reliable are the bounding boxes?

**Questions:**

Refer to weakness.

---

### Official Review · Reviewer_WkVS · 2024-11-04

**Soundness:** 2
**Presentation:** 2
**Contribution:** 2
**Rating:** 5
**Confidence:** 5

**Summary:**

The paper presents the MarineMaid dataset and benchmark designed for marine monitoring and understanding. It addresses the scarcity of labeled data in marine research and aims to improve the performance of object detection and vision-language understanding tasks for marine creatures. The dataset contains a wide range of marine creatures with detailed annotations, including 12,873 fine-grained instance-captioning pairs and 42,217 bounding boxes. The authors benchmark 14 state-of-the-art algorithms on this dataset to evaluate their performance and limitations.

**Strengths:**

1. The dataset provides detailed instance captions in addition to bounding box annotations, which is an advantage over existing marine datasets. These captions describe diverse biological traits of marine creatures, enhancing the understanding of the complex marine ecosystems. It includes 670 categories of marine objects, varying from Cephalopods, Crustaceans, Sharks, to Invertebrates, covering a wide range of marine life. The dataset incorporates a hierarchical taxonomy with 6 coarse-to-fine granularities, facilitating better classification and understanding of marine species.
2. The authors benchmark various marine visual understanding tasks, including close-set and open-vocabulary object detection, visual grounding, and instance captioning, using 14 state-of-the-art models. This provides a comprehensive assessment of the strengths and limitations of existing algorithms in the marine domain.

**Weaknesses:**

1. It would be better if the authors could propose improved algorithms or give more insightful analysis about the improvements. The experiments reveal that existing algorithms still face challenges in accurately detecting and understanding marine creatures. For instance, image-level VLMs often generate short and inaccurate captions, while region-level VLMs struggle to accurately localize specific marine
instances. General-purpose grounding algorithms also show a significant performance drop when applied to marine creature localization, indicating a gap in knowledge transfer from general to marine-specific tasks.
2. The authors used models such as generative AI and SAM to help construct the dataset, which raises concerns about the quality and diversity of the dataset. This severely limits the accuracy and performance of these models.
3. Since the dataset constructed by the author supports multiple tasks, is the effect of the multi-task joint pre-training model verified?

**Questions:**

1. Due to the fact that the datasets about marine science are not many, my question is that whether the author has conducted a complete literature review on marine datasets as shown in Table 1. Are existing marine understanding datasets included in Table 1? Since no new algorithms are proposed in this paper, I would expect this paper serves a better survey purpose, in order to publish in this leading conference.

**Details Of Ethics Concerns:**

The construction of this dataset involves  generative AI and SAM. Therefore, I have concerns about the ethics.

---

### Note · Authors · 2024-11-24

I have read and agree with the venue's withdrawal policy on behalf of myself and my co-authors.